# The Role of Genetic Factors in the Development of Acute Respiratory Viral Infection COVID-19: Predicting Severe Course and Outcomes

**DOI:** 10.3390/biomedicines10030549

**Published:** 2022-02-25

**Authors:** Mikhail M. Minashkin, Nataliya Y. Grigortsevich, Anna S. Kamaeva, Valeriya V. Barzanova, Alexey A. Traspov, Mikhail A. Godkov, Farkhad A. Ageev, Sergey S. Petrikov, Nataliya V. Pozdnyakova

**Affiliations:** 1Sistema BioTech LLC, 109235 Moscow, Russia; n.grigortsevich@sistemabiotech.ru (N.Y.G.); a.kamaeva@sistemabiotech.ru (A.S.K.); v.barzanova@sistemabiotech.ru (V.V.B.); a.traspov@sistemabiotech.ru (A.A.T.); n.pozdnyakova@sistemabiotech.ru (N.V.P.); 2Clinical Laboratory, N.V. Sklifosovsky Scientific Research Institute of Emergency Medicine, 129010 Moscow, Russia; mgokov@yandex.ru (M.A.G.); sklif@zdrav.mos.ru (S.S.P.); 3Clinical Laboratory, Moscow City Clinical Hospital #40, 129301 Moscow, Russia; ageev.fa@gmail.com

**Keywords:** COVID-19, single nucleotide polymorphism, genetic variant, SARS-CoV-2, heaviness, susceptibility, antiviral immunity

## Abstract

The aim of this study was to identify single nucleotide variants in genes associated with susceptibility to or severe outcomes of COVID-19. A total of 319 genomic DNA samples from patients with varying degrees of disease severity and 78 control DNA samples from people who had regular or prolonged contact with patients with COVID-19 but did not have clinical manifestations and/or antibodies to SARS-CoV-2. Seven SNPs were identified that were statistically associated with disease risk or severe course, rs1799864 in the *CCR2* gene (OR = 2.21), rs1990760 in the *IFIH1* gene (OR = 2.41), rs1800629 in the *TNF* gene (OR = 1.98), rs75603675 in the *TMPRSS2* gene (OR = 1.86), rs7842 in the *C3AR1* gene (OR = 2.08), rs179008 in the gene *TLR7* (OR = 1.85), rs324011 in the *C3AR1* gene (OR = 2.08), rs179008 in the *TLR7* gene (OR = 1.85), and rs324011 in the *STAT6* gene (OR = 1.84), as well as two variants associated with protection from COVID-19, rs744166 in the *STAT3* gene (OR = 0.36) and rs1898830 in the *TLR2* gene (OR = 0.47). The genotype in the region of these markers can be the criterion of the therapeutic approach for patients with COVID-19.

## 1. Introduction

Severe acute respiratory syndrome coronavirus (SARS-CoV-2) not only statistically leaves behind other viruses in terms of virulence but also negatively affects economic factors due to the quarantine and home restrictions that were introduced. In addition, people of middle and mature age, who make up the bulk of able-bodied people, including highly professional specialists, politicians, and medical workers contributing to the gross domestic product (GDP) of the Russian Federation, are the group most susceptible to infection and morbidity [1]. The study of genetic polymorphisms (single nucleotide polymorphism, SNP) that determine the manifestation of viral infection and coronavirus disease 2019 (COVID-19) was carried out in various categories of patients (different sexes, ages, concomitant disorders, etc.), both with pronounced clinical indicators and asymptomatic carriers of SARS-CoV-2 viral infection confirmed by PCR with real-time detection or a positive test for SARS-CoV-2 antibodies.

The purpose of our study was to establish relationships between variants in genes encoding key proteins of innate and adaptive immunity to COVID-19 with the severity of the disease and/or with the body’s resistance to the SARS-CoV-2 virus for the population of Moscow.

The study was carried out on genomic DNA samples from patients with a confirmed (PCR test and/or positive COVID-19 antibody test) COVID-19 diagnosis. An attempt was made to identify genetic markers (single nucleotide polymorphisms) associated with the course of the disease, as well as the risk of its occurrence. Previously published data from other populations have shown that the main risk factors for severe course and high mortality in COVID-19 disease are advanced age (>70 years), obesity, type 2 diabetes mellitus and, arterial hypertension [2,3,4,5]. Clinical studies of patients with COVID-19 have shown that the most frequent comorbidities are hypertension (56.6%), obesity (41.7%), and diabetes mellitus (33.8%) [6,7]. Previously, such data were not published on the population of Russia in general or the population of Moscow in particular.

In this study, polymorphic markers in genes, the activity of which was associated with the COVID-19 pathogenesis, were analyzed. When creating the research protocol, modern data obtained by scientists from different countries in the process of studying image-recognizing receptors and signaling pathways for immune response development were used [8,9,10]. When forming the list of markers, preference was given to genetic variants in the non-coding regions of genes (5′ and 3′ untranslated regions, as well as introns).

The analyzed genes can be divided into the following groups: (1) genes encoding members of the signaling pathways of the immune response mechanism to the SARS-CoV-2 virus (as well as to other viruses that cause respiratory infections) [11,12] and (2) genes encoding proteins responsible for the penetration of SARS-CoV-2 into human cells, including the angiotensin-converting enzyme ACE2 and the transmembrane protease TMPRSS2 [13,14,15].

We analyzed the genotype in the region of 32 polymorphic markers localized in 29 genes encoding proteins of the above-mentioned signaling pathways.

Of these, nine SNPs showed a statistically significant association with the severity of COVID-19 disease, of which two were protective and seven were risk factors. A protective character for rs744166 (GG genotype) was identified in *STAT3* (OR = 0.37, 95% CI = 0.19–0.72, *p* = 0.0025). Protection was also identified in relation to disease severity for rs1898830 (GG genotype) in *TLR2* (OR = 0.47, 95% CI = 0.25–0.9, *p* = 0.017).

The following markers appeared as risk factors: rs1799864 in *CCR2* (allele A, OR = 2.21, 95% CI 1.12–4.39, *p* = 0.015)*,* rs1990760 in *IFIH1* (genotype CC, OR = 2.41, 95% CI = 1.11–5.26, *p* = 0.016), rs1800629 in *TNF* (allele A, OR = 1.98, 95% CI = 1.23–3.18, *p* = 0.0042)*,* rs75603675 in *TMPRSS2* (allele A, OR = 1.86, 95% CI = 1.20–2.89, *p* = 0.005)*,* rs7842 in *C3AR1* (allele C, OR = 2.08, 95% CI = 1.25–3.46, *p* = 0.0042)*,* rs179008 in *TLR7* (allele T, OR = 1.85, 95%CI = 1.03–3.32, *p* = 0.036), and rs324011 in the *STAT6* (allele T, OR = 1.84, 95%CI = 1.01–3.36, *p* = 0.041).

The identified genetic determinants provide direction both for identifying the risk groups for severe COVID-19 and for possible therapeutic approaches.

## 2. Materials and Methods

### 2.1. Clinical and Demographic Characteristics of the Studied Patients

This study included patients of the intensive care unit (ICU) of the N.V.Sklifosovsky Scientific Research Institute of Emergency Medicine, Clinical Hospital No. 1 of Medsi, City Clinical Hospital No. 40, as well as employees of AFK Sistema and Sistema BioTech Laboratory. The study was conducted according to the guidelines of the Declaration of Helsinki and approved by the Ethics Committee of N.V. Sklifosovsky Scientific Research Institute of First Aid (meeting minutes of Committee No. 1–21 dated 18 February 2021). The total sample size was 397 people, including 319 patients with a positive PCR test for SARS-CoV-2 and a diagnosis of COVID-19. PCR tests were carried out on RNA extracted from oropharyngeal swabs. According to CT-SS (computed tomography severity score), the patients were divided into three subgroups: the first subgroup (mild, 99 persons) included asymptomatic circulators (no symptoms) and patients with mild pulmonary involvement (CT-1); the second subgroup (medium, 65 persons) included patients with moderate pulmonary involvement (CT-2), and the third subgroup (severe, 155 persons) included patients with severe (CT-3) and extremely severe (CT-4) pulmonary involvement.

The severity of the COVID-19 disease was determined by chest computer tomography (CT) scan [16].

The control group consisted of 78 persons who had long-term contacts with patients, tested negative for regular PCR testing for SARS-CoV-2, and had no antibodies to SARS-CoV-2, including medical specialists of the departments for COVID-19 patients and family members of patients who had COVID-19. The demographic data are presented in Table 1 and Table 2, and the full phenotypic data and medical histories of the patients are presented in Appendix A.

### 2.2. Genomic DNA Isolation

EDTA-stabilized venous blood was used as a biological material for genomic DNA isolation.

DNA was isolated using a DiaGene whole blood DNA extraction kit (Dia-M, Moscow, Russia). The purity of the isolated DNA was tested with the NanoDrop OneC spectrophotometer (Themo FS, Waltham, MA, USA). The A260/280 ratio ranged from 1.8 to 1.91, and the A260/230 ratio ranged from 1.62 to 2.28. The DNA concentration was measured using a dsDNA BR kit on a Qubit Flex fluorometer (Thermo FS, Waltham, MA, USA). The concentration ranged from 15 to 300 ng/μL. The concentration of all DNA samples was adjusted to 2 ng/μL.

### 2.3. DNA Genotyping

Genotyping of DNA in the field of the polymorphic markers under study was carried out by real-time PCR with fluorescence detection using kits for the genotyping analysis of TaqMan SNP (Thermo FS, Waltham, MA, USA). PCR was performed on a CFX96 amplifier (BioRad, Hercules, CA, USA).

A number of markers were genotyped to confirm the sequence by Sanger sequencing. The primers were developed using PrimerBlast (https://www.ncbi.nlm.nih.gov/tools/primer-blast/ (accessed on 24 September 2020)). A BigDye Terminator v3.1 cycle sequencing reaction used the BigDye cycle sequencing kit (Thermo FS, Waltham, MA, USA); sequencing was performed on the Genetic Analyzer 3500 (Thermo FS, Waltham, MA, USA).

### 2.4. Statistical Processing of the Obtained Data

To calculate the statistical significance of the differences in the frequencies of occurrence, the exact two-tailed Fisher test and the chi-squared test were used. Calculations were made for both the minor allele (AA vs. Aa + aa, the dominant model) and the homozygous genotypes for the minor allele (AA + Aa vs. aa, recessive model). Further, for groups where significant differences were found, odds ratios (OR), a 95% confidence interval, and a criterion of statistical significance, *p,* were calculated. Calculations were carried out using the MATLAB 2010b resource and the SNPStats software (https://www.snpstats.net/ (accessed on 20 September 2021), which is designed to identify links between single-nucleotide polymorphisms and disease risk. To avoid the effect of multiple comparisons for the OR values of statistically significant polymorphisms, a Holm–Bonferroni correction was made.

## 3. Results

### 3.1. Allele Frequency and Population Analysis

Based on the results of the analysis of genotyping results, the frequencies of occurrence of alleles and genotypes for the analyzed polymorphisms were obtained. For some rare polymorphisms (e.g., rs61735794 in the *TMPRSS2*), the sample size was insufficient to identify the number of individuals with a minor allele needed for statistical calculations. These polymorphisms have been excluded from the statistical processing and no data are given on them. The frequencies of occurrence of the minor alleles of polymorphisms selected for statistical processing are shown in Table 3. All genotyping data are presented in Table 1 in the Additional Data section.

According to Table 3, most minor alleles have a frequency of occurrence similar to the European one. Three markers showed a frequency close to the frequency of the Asian population—rs1799864 in *CCR2*, rs1898030 in *TLR2*, and rs352162 in *TLR9*. The frequency of the minor alleles for the two markers differed from both European and Asian data: rs2074192 in *ACE2* and rs1800925 in *IL13*. There is no data for rs7842 in *C3AR1* in database 1000 Genomes.

### 3.2. Statistical Analysis

To calculate the statistical significance of the difference in the frequencies of occurrence, the exact two-tailed Fisher test and the chi-squared test were used. The statistical calculations were carried out according to the dominant model (identification of the risk allele) and the recessive model (identification of the risk genotype). Furthermore, for groups where significant differences were found, odds ratios (OR), 95% confidence interval, and the criterion of statistical significance, *p,* were calculated. For the convenience of calculating the difference between the groups, three groups of severity were determined, taking into account each polymorphism: (1) control, from mild to moderate, and severe; (2) control, mild, moderate and severe; (3) control in all cases. All genotyping data are presented in Table 1 in the Additional Data section.

Thus, for all polymorphisms, three options (depending on the division of patients into groups) were made for calculating the statistical significance using two models: dominant and recessive.

The first option was intended to assess the risk of developing a mild form into a more severe one; the incidence was compared in healthy, asymptomatic, and mild patients on the one hand (group A1) and in patients of the moderate and severe form on the other (group A2).

The second option was intended to assess the risk of serious illness. In this case, the frequencies were compared in the group “healthy + light + medium” (group B1) and in the group of severe (group B2).

The third option was to assess the risk of infection with SARS-CoV-2; the frequencies were compared in the healthy group (group C1) and the group that included all patients (group C2).

The analysis of the Hardy–Weinberg equilibrium of the sample including 397 subjects has been performed for these SNP markers, as well as rs 12252 in the IFITM3 gene. Although this polymorphism did not show statistical significance, we included it into further consideration due to its potentially clinical significance. Six SNPs did not deviate from the Hardy–Weinberg equilibrium, three slightly deviated, and one (rs179008) substantially deviated (data shown in Table 4 and Appendix A). We propose that deviation from the HWE occurred due to an insufficient control subgroup.

In total, nine polymorphic markers were identified, the frequencies of which showed statistically significant differences between certain groups (see Table 5, Table 6 and Table 7 and Appendix A).

Since the experiment tested a large number of hypotheses on the same data set, due to the effect of multiple comparisons the Holm-Bonferroni correction was made to avoid type I errors. All hypotheses showed their significance.

For the two markers, the protective factor was the homozygous genotype for a minor allele: rs744166 in the *STAT3* gene for susceptibility to COVID-19 and rs1898830 for the *TLR2* gene for the progression of the disease to severe form.

For seven markers, the presence of a small allele or homozygous genotype for a minor allele was somehow significantly associated with increased susceptibility to disease or with an increased risk of severe disease. These are rs1800629 in the *TNF,* rs75603675 in the *TMPRSS2,* rs1799864 in the *CCR2,* rs1990760 in the *IFIH1* gene, rs7842 in the *C3AR1* gene, rs324011 in the *STAT6* gene, and rs179008 in the *TLR7* gene.

Evaluated OR, 95% CI, and *p*-values for statistically significant SNPs are presented in Table 5, Table 6 and Table 7 below. Statistical data for all studied SNPs are presented in Appendix A.

We also analyzed our samples including the polymorphisms passing the statistical significance level of 0.05. The analysis was carried out in the Statsoft Statistica 12 package. To analyze the required sample size at a given power of 80, we used a 2-sample independent *t*-test to compare 2 averages between 2 populations (the null hypothesis μ1 = μ2). These data are presented in Appendix A.

All identified markers are risk factors for both susceptibility to the disease and for the development of the disease into a more severe form, but statistical significance was achieved only for the variants shown in the tables above.

Mention should also be made of the rs12252 polymorphism in the *IFITM3* gene, for which the minor C allele was associated with susceptibility to COVID-19. From the obtained data, it follows that the CK genotype homozygous for the minor allele is a risk factor for increasing the severity of the disease (OR = 3.26), but statistical significance was not achieved due to the rare appearance of the C allele in the European population (just over 4% according to 1000 Genomes).

## 4. Discussion

According to information from key literature reference books for August 2020, other scientists found that minor alleles of polymorphisms in the genes *CCL2, IFIH1, MBL2, IFITM3,* and other genes were associated with SARS-CoV, as well as with other viral respiratory infections [17,18,19]. To date, studies have not provided a holistic understanding of the molecular genetic mechanisms underlying a person’s susceptibility to one or another manifestation, outcome, and/or complications of COVID-19 disease due to insufficient data and the novelty of SARS-CoV-2.

In the pathogenesis of COVID-19 and the development of such a terrible complication as the “cytokine storm”, several types of immune cells of the human body and the most important signaling pathways are involved.

The proteins involved in COVID-19 pathogenic pathways (and the genes encoding them) can be divided as follows, depending on their function.
Recognition of surface viral antigens (image-recognizing receptors, ACE-2).Interaction of viral particles with specific structures on the surface of the cell and penetration of viral particles into the cell.Intracellular recognition of viral RNA.Launching intracellular signaling pathways and activating the immune system, including with the formation of a “cytokine storm” and complement activation.Participation in the mechanisms of pathogenesis of complications of COVID-19: damage to blood vessels and pulmonary epithelium, the likelihood of developing sepsis, septic shock, and multiple organ failure.

Table 3 shows the key genes of these processes and their polymorphisms (genetic variants) that may influence the activity of these genes or the proteins encoded by these genes (as of August 2020). These genes can be divided into four groups. Three of them include genes of signaling pathways connected with the body’s immune response to the RNA viruses (including SARS-CoV-2), and the fourth group consists of genes connected with the specific penetration of SARS-CoV-2 into human cells.

After mathematical processing, only nine polymorphisms from the above list showed a statistically significant association with susceptibility to COVID-19 and/or its severe course. This is not to say that there is no link between other markers and the disease. An increase in sample size is expected to increase the number of significant markers, as a number of them showed a tendency to turn into risk factors, for example, rs721917 in the *SFTPD* gene (a component of the pulmonary surfactant that protects the lungs from microorganisms) or rs2227956 in the *HSPA1A* gene (heat shock protein).

Notably, none of the polymorphic markers studied in the *ACE2* gene, whose product is the main receptor for SARS-CoV-2, showed any association with either susceptibility to COVID-19 or its severity (data not shown). The selection of markers in this gene was based on the assumption of the relationship between intron mutations and the level of expression of this gene. The absence of a link between genotype and susceptibility to the disease does not mean that the activity of the gene does not affect the susceptibility of the organism to coronavirus infection. This question needs to be studied on the basis of data relating not so much to mutations in the *ACE2,* but to the level of its transcription.

### 4.1. TNF

TNF-α is a multifunctional pro-inflammatory cytokine that belongs to the superfamily of tumor necrosis factors (TNF). This cytokine is mainly secreted by macrophages. It can bind to its TNFRSF1A/TNFR1 and TNFRSF1B/TNFBR receptors and perform its functions through them. This cytokine is involved in the regulation of a wide range of biological processes, including cell proliferation, differentiation, apoptosis, lipid metabolism, and coagulation.

The role of the rs1800629 polymorphism in the *TNF* gene has been widely discussed due to many inflammatory diseases, but there are virtually no publications on its role in the susceptibility and/or severity of COVID-19.

Saleh A. et al. tested 900 patients with COVID-19 infection and 184 people in the control group (medical workers who had contact with COVID-19 patients between April and July 2020) for polymorphism of the *TNF* G-308A promoter [20]. Different genotypes of *TNF* G-308A (rs1800629) were compared to the disease’s severity and prognosis. The polymorphism located at -308 before the initiation point of transcription ireplaces G (TNF1) with A (TNF2), which is responsible for a 6–9-fold increase in TNF transcription in vitro and higher plasma levels of TNF in vivo. After genotype testing, a statistically significant difference was found between patients and the control group in terms of genotype distribution, where the A allele is more pronounced in patients compared to the control with *p* = 0.005. A total of 420 (46.7%) patients had AA, 288 (32.0%) had HA, and 192 (21.3%) had GG versus 60 (32.6%), 40 (21.7%), and 84 (45.7%) in the control group, respectively. This indicates that people carrying the A allele (AA and GA) are more susceptible to the disease. In addition, in the AA genotype, the disease was severe in 336 (80.0%) cases compared to 120 (41.7%) cases in the GA genotype and never in the GG genotype with *p* = 0.001. The time before the first and second negative swabs was significantly longer in the AA genotype. Mechanical ventilation was required in 120 cases with the AA genotype compared to 12 cases with the GA genotype and in no cases with the GG genotype [20].

This data suggests that allele A is associated with a more aggressive disease, and this may be due to variations in serum TNF-α levels.

Ding et al. demonstrated an association between the A allele rs1800629 in *TNF* and the G(C) allele rs1800796 in *IL6* and a higher susceptibility to acute respiratory distress syndrome (ARDS) [21].

### 4.2. TMPRSS2

The *TMPRSS2* gene encodes a transmembrane protein from the serin proteinase family. From the point of view of the pathogenesis of viral infections, this protein is of considerable interest, as it breaks down the glycoproteins of viruses that cause respiratory infections, such as influenza [22], parainfluenza [23], SARS, and MERS [24], ensuring their penetration into cells.

Moreover, this protein, together with ACE2, is involved in the penetration of the SARS-CoV-2 virus into human cells, breaking down the viral S-protein [14]. In addition, S1 fragments cleaved by TMPRSS2 from the viral S protein bind virus-neutralizing antibodies and inactivate them [15]. Because TMPRSS2 plays an important role in the pathogenesis of COVID-19, the activity of the gene encoding this protein is associated with susceptibility to the virus and the severity of the disease.

The role of structural and regulatory variations of TMPRSS2 in susceptibility to COVID-19 is being actively studied. There is evidence that genetic variants potentially associated with conformational changes and low expression of TMPRSS2 (and ACE2) are less common in African populations [13,25] and more common in southeast Asia and the far east.

The investigated polymorphism rs75603675 (23 G > T, Gly8Val), along with rs12329760 (589 G > A, Val197Met), is theoretically considered responsible for changes in the interaction of TMPRSS2 with the S-protein of the virus and ACE2 [26,27]. The results of mathematical modeling indicate that the replacement of glycine, which is the smallest amino acid, with a hydrophobic valine with a long side chain should lead to a decrease in the functional activity of proteinase due to a decrease in the flexibility of the peptide and an increase in hydrophobicity. On the other hand, this substitution increases the stability of the model protein. It is believed that such substitution should lead to a decrease in the ability of TMPRSS2 to bind the S-protein [14].

However, the obtained data indicate that circulators of the minor T allele are more susceptible to infection with SARS-CoV-2 (OR = 1.71, 95% CI 1.03–2. 83, *p* = 0.039) and have a higher risk of developing a mild form of COVID-19 into a medium (OR = 1.83, 95% CI 1.21–2.78, *p* = 0.0043) or severe one (OR = 1.86, 95% CI 1.2–2.89, *p* = 0.005).

TMPRSS2 is considered a potential target for COVID-19 therapy [28]. Hoffmann et al. found that a serine protease inhibitor, the camouflage mesylate, partially blocks TMPRSS2-ACE2-mediated SARS-CoV-2 input. Similarly, nafamhostate mesylate inhibited TMPRSS2-ACE2-mediated fusion of the SARS-CoV-2 envelope with the plasma membrane and then showed a 10-times greater efficacy than kamostat mesylate in preventing SARS-CoV-2 from entering cells [28].

### 4.3. STAT3

The *STAT3* gene encodes a transcription activator protein from the STAT family. It acts as an expression regulator for a number of genes in response to cytokines and growth factors. In addition, STAT3 is involved in regulating the body’s response to viral and bacterial infections since the interaction of IL-6 and IL-10 with their respective receptors triggers the phosphorylation process of STAT3. Interleukin-6 levels are associated with the severity of the infectious disease and the risk of complications [29].

Interleukin-27 triggers the phosphorylation of STAT3 in various types of immune cells [30] and induces the production of anti-inflammatory IL-10 by regulatory T cells [31]. Thus, the activity of STAT3 is associated with inhibition of the inflammatory process. The connection of the genotype rs744166 in the 2 intron of the *STAT3* gene is associated with a number of chronic inflammatory diseases, including Crohn’s disease [32], disseminated sclerosis [33,34], and cancer [35,36,37,38]. At the same time, the small allele G has a protective effect against Crohn’s disease (OR = 0.83) [39].

In [40], it was shown that polymorphisms in the regulatory region of the *STAT3* gene, which reduce the level of its expression, lead to an increase in the effect of alpha-interferon (shown in a model with tumor cells where reduced expression of STAT3 induced sensitivity to interferon).

The association of variants of the *STAT3* gene with susceptibility to infectious diseases was also shown.

STAT3 is seen as one of the key elements of cytokine storm pathogenesis and other complications in COVID-19 that target IL-6 [41]. The production of IL-6 by epithelial cells in COVID-19 leads to the activation of STAT3 [42]; therefore, gene variants that reduce expression may have a protective effect against SARS-Cov-2. Since STAT3 is a regulator of the activity of many genes responsible for the development of an inadequate immune response to viral infection, it, unlike other elements of the signaling pathway involving STAT3, represents a more attractive target for the therapeutic prevention against the cytokine storm.

### 4.4. STAT6

The signal converter and transcription activator 6 (STAT6) can function as a signaling molecule and as a transcription factor. STAT6 is activated by intracellular foreign nucleic acids, which lead to the activation of innate immunity [43]. Activated in this way, STAT6 regulates a specific set of genes needed to recruit different immune cells to the site of infection. STAT6 is the signaling mediator of IL4 and IL13 and promotes the anti-inflammatory process by inducing the development of Th2 lymphocytes and M2 macrophages. Canonical activation of STAT6 in the IL4 and IL13 signaling pathways is mediated by the tyrosine kinase JAK [44]. Virus-induced activation of STAT6 has been found to be independent of cytokines and JAK [45]. TBK1 kinase has been shown to phosphorylate STAT6, which in turn induces dimerization and the translocation of STAT6 into the nucleus, resulting in the induction of the CCL2, CCL20, and CCL26 chemokines in an IFN-independent manner, and neither NF-κB nor IRF3 are critical for the virus-induced transmission of STAT6 signals, suggesting that STAT6-mediated signaling bifurcation occurs at an early stage [45].

RNA virus infection triggers the activation of STAT6 through the interaction of STING, TBK1, and the MAVS adapter protein. Increased expression of this protein, a member of the family of transcription factors in rectal and/or breast cancer, is associated with cancer cell proliferation, increased malignancy, and poor prognosis. Salguero-Aranda C. et al. studied the inactivation of STAT6 in vitro using small interfering RNAs (siRNAs) specific to *STAT6*. The suppression of *STAT6* also significantly induced apoptotic events. SiRNA STAT6 sequences have been demonstrated to be able to inhibit the proliferation and induce apoptosis of colorectal HT-29 cancer cells and ZR-75-1 breast cancer cells, halving the number of cancer cells in a short period of time. Thus, siRNA STAT6 sequences represent a potential treatment for colon and breast cancer with a high degree of STAT6, as well as a large number of other malignant tumors expressing STAT6 [45,46].

### 4.5. TLR

The toll-like receptor family is an important component of innate immunity that is responsible for the interaction of components of viral particles (proteins or genetic material) with cellular structures. This family includes the proteins TLR2, TLR3, TLR4, TLR7, TLR9. They play a fundamental role in recognizing pathogens and activating innate immunity. Toll-like receptors usually exist as homodimers (heterodimers have been reported) and are found on immune cells, macrophages, B lymphocytes, and mast cells.

Activation of TLR-dependent signaling pathways leading to the secretion of pro-inflammatory cytokines (interleukin-1, interleukin-6, tumor necrosis factor-α, and interferon type 1) occurs when various infectious agents, including SARS-CoV-2, enter the body. Therefore, the analysis of the genotypes of *TLR2, TLR3, TLR4, TLR6, TLR7, TLR8,* and *TLR9* is important for the study of COVID-19.

TLR2/6 and 4 are localized on the cell membrane, while TLR3, TLR7/8 and 9 are localized on the surface of the endosomes. TLR activation of two major adaptive proteins, the products of the MYD88 and TICAM1 genes, leads to the production of pro-inflammatory cytokines and interferon-1. Activation of TICAM1 triggers a cascade of successive activations of other genes, *TICAM1, TRAF3, IRF3*, which ultimately leads to the production of interferon-1. MYD88 induces activation of TRAF6, as well as the TLR4 pathways, but leads to activation of IRAK4 and indirectly TRAF6 and TRAF3 in the TLR7/9 and TLR9 pathways. Activation of NF-kB signaling via TRAF6 increases the production of pro-inflammatory cytokines, and activation of *IRF7* also leads to the production of interferon-1 [47].

The rs1898830 variant in the *TLR2* intron gene has previously been significantly associated with severe respiratory syncytial infection and bronchiolitis in neonates [48]. Its functional significance is still unclear. However, it has been suggested that the small G allele may alter the expression or functionality of *TLR2,* resulting in an increase in the number and/or function of macrophages and endothelial cells, thereby enhancing the innate immune response, which is also involved in the pathophysiology of atherosclerosis [49,50]. The association of this variant with COVID-19 has not been studied. Studies have revealed the protective nature of the minor genotype of GG, both in terms of susceptibility to COVID-19 and in terms of the severity of the disease. However, statistical significance was shown only for the risk of mild or moderate becoming severe (OR = 0.47, 95% CI = 0.25–0.90, *p* = 0.017). The assumption that polymorphism is related to the level of expression may explain the protective nature of the GG genotype, but this polymorphism needs further investigation using differential gene expression analysis techniques.

The circulars of the minor T allele in rs179008 of the *TLR7* gene have significantly lower expression of the interleukin-29 (IL-29) gene, also called interferon lambda. This is a risk factor for increased susceptibility and more severe viral infections [51].

### 4.6. C3AR1

*C3AR1* is the genome encoding the inflammatory anaphylatoxin C3a receptor, which is released when the complement is activated. C3a binding by this receptor activates chemotaxis, granular enzyme release, superoxide anion production, and bacterial opsonization.

It is now widely known that lipid disorders, obesity, and type 2 diabetes are risk factors for severe COVID-19, and adipose tissue is considered not just a store of fat, but an endocrine organ that secretes many metabolically active adipokines and hormones derived from adipose tissue. The link between obesity and insulin resistance is complex, causing a number of metabolic disorders, including type 2 diabetes (T2DM), dyslipidemia, and bleeding disorders [52].

Adipokine adipsin plays a beneficial role in maintaining the function of β cells. Animals genetically deprived of adipsin have glucose intolerance due to insulinopenia; isolated islets of these mice have reduced insulin secretion when stimulated by glucose [53]. It was later found that adipsin is a complement factor D [54]. Lo et al. suggest that adipsin stimulates insulin secretion through the action of C3a on its C3aR1 receptor [55]. In a study of GWAS among the Chinese population, variant 3′ UTR *C3AR1* (rs7842 A/G) was identified in the eQTL analysis and was associated with *C3AR1* expression levels and coronary heart disease. It was found that the rs7842 variant located in the 3′ UTR of the *C3AR1* gene was associated with increased expression of C3AR1 (*p* = 5.47 × 10^−5^). For rs7842, the average relative expression levels of *C3AR1* were 1.42 ± 0.24 for 14 circulators of the GG genotype, 1.10 ± 0.23 for 67 circulators of the AG genotype, and 0.92 ± 0.18 for 185 circulators of the AA genotype. A statistical analysis using general linear modeling showed that the minor G allele rs7842 was significantly associated with a higher level of C3AR1 mRNA expression in the dominant model with standardized coefficients (β) of 0.35 (mean expression: 1.13 ± 0.30 for GG + GA versus 0.92 ± 0.18 for AA, *p* = 4.07 × 10^−9^) [56]. In this study, G small allele circulators showed higher levels of *C3AR1* expression in white blood cells. However, C3AR1-related indicators, such as C3a and adipsin were not considered in this study.

Yan B, et al. found increased C3 expression in alveolar type 2 pneumocyte cells, which are the main targets for virus penetration. Increased expression was observed more often in covid-19 patients than in uninfected donors, suggesting that cell infection induces transcription of the C3 gene. Additional autopsy data showed that fewer patients with COVID-19 infection had a positive correlation between increased C3 mRNA expression and the SARS-CoV-2 viral load [57].

High expression of C3AR1, the gene encoding the proteins C3aR and CD46, was observed in lymphoid cells. The expression of genes regulated by CD46 was significantly higher in the lymphoid cells from the lungs of patients with more severe COVID-19. Genes regulated by C3aR are significantly more strongly expressed in monocyte/macrophage cells in patients infected with COVID-19 [57].

### 4.7. CCR2

The function of chemokines is to regulate the activity of leukocytes during the immune response or inflammation [58]. Chemokines work through receptors, one of which is CCR2. This is the main chemokine receptor on the surface of monocytes, T-lymphocytes, and macrophages [58]. The ligands for this receptor are proteins from the family of monocytic chemoattractants, MCP-1 (CCL-2), MCP-2 (CCL-8), and MCP-3 (CCL-7).

Increased production of the CCL2 chemokine has previously been shown to be associated with severe acute respiratory distress syndrome (ARDS) caused by coronavirus 2 (SARS-Cov) [18]. CCL2 is an important chemokine associated with the severity of COVID-19. It is activated at the early stage of an infectious disease, and in the later stages in fatal cases its concentration rises sharply. In the lungs, CCL2 is mainly produced by alveolar macrophages, T lymphocytes, and endothelial cells, while its related CCR2 receptor is mainly expressed on monocytes and T lymphocytes [59]. In addition, the presence of CCR2-bearing blood monocytes enhances neutrophil accumulation, dramatically reflecting the cooperation and coordination between monocytes and neutrophils in leukocyte efflux during pneumonia. In addition, CCL2 has been reported to increase procollagen synthesis by fibroblasts. Together, these CCL2 functions can lead to fibroproliferative complications in ARDS.

One review [59] was devoted to the place of chemokines and their receptors in the pathogenesis of COVID-19 and the relationship between the level of their expression and the severity and prognosis of the disease. Comparing the chemokine profile of SARS-CoV-2 with SARS-CoV and MERS-CoV, it was concluded that CXCL8, CXCL10, and CCL2 make a decisive contribution to the pathogenesis of the disease in all three coronaviruses.

An earlier study after the SARS-Cov epidemic [18] showed that the G-2518A *codon variants CCL2* (rs1024611) and *MBL54* (rs1800450) had a significant cumulative effect on increased susceptibility to SARS-CoV infection.

The rs1799864 polymorphism is a G/A point mutation at position 8974 (NG_021428,1) of the *CCR2* gene, leading to the replacement of the amino acids Val64Ile. This polymorphism is associated with the susceptibility and severity of viral infections, such as HIV infection [60] and hepatitis C [61,62]. 

### 4.8. IFIH1

The *IFIH1* gene encodes the MDA5 helicase receptor, bound to RIG-like receptors, which plays an important role in detecting viral double-stranded RNA and activating the antiviral cascade. This is an intracellular viral RNA sensor that triggers an innate immune response. When ligands bind, it binds to a mitochondrial antiviral signaling protein (MAVS/IPS1), which triggers two signaling pathways. First, through the kinase system of the IKK, it activates the transcription factor NF-kB, which causes cell activation and the synthesis of a number of pro-inflammatory cytokines. The second, through the adapter of the TRAF3 protein and the protein kinase TBK1, activates two transcription factors, IRF3 and IRF7, inducing the synthesis of interferons beta and alpha, respectively [63]. Mutations in this gene are significantly associated with various autoimmune diseases (systemic lupus erythematosus, rheumatoid arthritis, multiple sclerosis, and latent autoimmune diabetes LADA) and type 1 diabetes mellitus [64]. In relation to SARS-Cov-2, the MDA5 receptor is a receptor that recognizes virus RNA and triggers the production of interferon as the first line of antiviral defense [65].

At the functional level, Gorman et al. recently studied the effects on viral sensitivity and autoimmune pathogenesis of rs1990760, an erroneous variant of *IFIH1* that is associated with multiple autoimmune diseases (type 1 diabetes mellitus, systemic lupus erythematosus, vitiligo, autoimmune thyroid disease, etc.) [66]. They showed that the allele that provides the best protection against viral infection also enhances autoimmune reactions against its own RNA [67,68]. Together these findings highlight the key role of innate immune recognition and activation in the complex balance between host defense, inflammation, and autoimmunity. Bastard et al. identified people with high titers of neutralizing autoantibodies against IFN-α2 type I and IFN-ω in about 10% of patients with severe COVID-19 pneumonia. These autoantibodies were not detected in infected people who had no symptoms or who had a milder phenotype, or in healthy people. About 10% of patients with life-threatening COVID-19 pneumonia had neutralizing autoantibodies against type I interferons [69].

The circulators of genotype rs1990760 TT *IFIH1* had lower serum levels of IL-18 (*p* < 0.001) and granzyme B (*p* < 0.001) than the circulators of the CC and CT genotypes. Patients with systemic lupus erythematosus of the Chinese genotype circulators of rs 1990760 *IFIH1* had higher anti-dsDNA-positive results than the circulators of the CC and TT genotypes [68].

### 4.9. IFITM3

*IFITM3* (interferon-induced transmembrane protein 3) is a gene encoding a protein that helps build immunity to influenza A H1N1 virus, West Nile virus, and dengue virus. It inhibits the penetration of viruses into the cytoplasm of the host cell, preventing the fusion of viruses with cholesterol-depleted endosomes. It is active against several viruses, including the influenza A virus, SARS coronavirus (SARS-CoV), Marburg virus (MARV), Ebola virus (EBOV), dengue virus (DNV), West Nile virus (WNV), human immunodeficiency virus type 1 (HIV-1), and vesicular stomatitis virus (VSV). It may suppress viral penetration mediated by influenza hemagglutinin protein, viral penetration mediated by GP1,2 MARV and EBOV, viral penetration mediated by SARS-CoV protein, and viral penetration mediated by VSV G protein.

The allele in the polymorphic embodiment rs12252 (G) encodes an isoform of IFITM3 (known as Δ21 IFITM3), which is missing 21 amino acids in the N-terminal region of the protein. The frequency of rs12252(G) alleles varies widely between Caucasian and Asian patients; while there are very low levels in Caucasians, up to 25–50% of all Japanese or Chinese may have the rs12252 (G/G) genotype. A study conducted by a Han Chinese man found that the rs12252 genotype (G/G) is estimated to carry six times the risk of severe infection than genotypes A/G and A/A. The G allele is a hypothetical allele of the risk of severe infection [69]. Similar results were obtained for the Korean population for influenza type A [70]. A similar study conducted in the population of the Asturias region of northern Spain– confirmed the findings of the Asian scientists. The C(G) allele in rs12252 *IFITM3* was found to be a risk factor for hospitalization with COVID-19 in the European population [71]. The degree of association was lower in the Chinese population, but with a much higher risk allele rate [69].

## 5. Conclusions

In this work, an attempt was made to identify the genetic determinants of morbidity and risk of severe course for the Russian (Moscow) population of respiratory viral infections. Genetic polymorphic variants in genes, according to foreign literature data, involved in the pathogenesis of COVID-19 were investigated.

A group of statistically significant genetic variants associated with COVID-19 disease was identified. At the same time, it is interesting that almost all these genetic variants are located in genes that are somehow responsible for innate (primary) immunity, including *CCR2*, *IFIH1*, *TMPRSS2*, *TNF*, *C3AR1*, *STAT3*, *STAT6*, *TLR2,* and *TLR7.* Minor homozygous variants in the *STAT3* and *TLR2* genes are protective in terms of both susceptibility and risk of exacerbation of the disease and severity of the course.

The findings suggest that genotyping of these polymorphic markers by standard methods (PCR, capillary sequencing, or pyrosequencing) as part of clinical laboratory diagnosis may become a routine prognostic test for the development of an asymptomatic form into a form with clinical manifestations and/or severe course. Since the genetic markers under study are associated not only with infection with the SARS-CoV-2 virus but are also non-specific for the immune response to viral agents, genotyping of these markers can be further used to predict the course of other respiratory viral infections. The COVID-19 pandemic in Russia alone, according to the Russian government (https://economy.gov.ru/material/file/f636b757cf29f285a839ce95b2d1123f/Plan.pdf; accessed on 10 September 2021), has affected about 6.7 million people and caused both a drop in GDP and household incomes and an increase in health care costs (due to the high costs of treating COVID-19 patients). It is assumed that the prognostic identification of risk factors for severe COVID-19 disease and, theoretically, other respiratory viral infections can reduce mortality, the number of complications, the severity of complications, and the cost of therapy due to the earlier start of treatment with the choice of individually selected targeted drugs. This was reflected in the discovery of the relationship between markers and violations of certain processes in the production of endogenous interferons, sensitivity to glucocorticosteroid therapy, early use of monoclonal antibodies, and convalescent plasma.

Genetic predisposition to infectious diseases is still one of the most promising areas for study. Since the early 50s, many monogenic primary immunodeficiencies have been discovered, and at the population level, a number of large genes have been discovered for a number of infectious diseases. For a number of mutations, their evolutionary protective value is justified, and some of them celebrated the 2000th anniversary of their formation in human DNA. Long-term studies of adaptation in children also show that the predisposition to infectious diseases is inherited more strongly than in other diseases associated with the likely impact of environmental factors, such as tumors [72].

In a number of studies, it was found that the concordance coefficient for infectious diseases is higher in monozygotic twins than in dizygotic twins, which also indicates the relationship between the genetic background and susceptibility to infections [72]. Thus, based on the statement about the genetic determinism of infectious diseases and their polygenicity, it is necessary to study genetic determinism to determine the hierarchy of genetic markers for both an individual and a population as a whole. Since the beginning of the coronavirus epidemic, the need to predict morbidity and mortality and assess the economic components and structures of medical care during the epidemic has been shown.

An attempt to detect such determinism was made in our research paper on the search for differences in the genotypes of patients with different courses of Covid-19.

## Figures and Tables

**Table 1 biomedicines-10-00549-t001:** Gender and severity of COVID-19 disease.

Severity	Total	Men	Women
Control group	78	41 (52.6%)	37 (47.4%)
Easy	99	55 (55.5%)	44 (44.5%)
Tolerant	65	32 (55.4%)	24 (44.6%)
Strong	155	104 (67.1%)	51 (32.9%)

**Table 2 biomedicines-10-00549-t002:** Distribution by age and severity of the disease.

Age	21–39	40–49	50–59	60–69	70–79	≥ 80
Soft (99)	10 (9.9%)	15 (15.2%)	24 (24.2%)	23 (23.3%)	19 (19.2%)	8 (8.1%)
Intermediate (65)	8 (12.3%)	7 (10.8%)	10 (15.4%)	14 (21.5%)	12 (18.5%)	14 (21.5%)
Heavyweight (123)	6 (4.9%)	17 (13.8%)	32 (26%)	41 (33.3%)	25 (20.3%)	2(1.6%)
Extremely heavy (32)	-	3 (9.4%)	8 (25%)	10(31.3%)	6 (18.8%)	5 (15.6%)

**Table 3 biomedicines-10-00549-t003:** Polymorphic markers, their localization, and frequency of occurrence.

Gene	Gene Function	Polymorphism	Genotype ^1^	Localization of Polymorphism	Frequency of Small Alleles (Data from 1000 Genomes)			Frequency of Occurrence of the Minor Allele in the Entire Sample
					EUR ^2^	EAC ^2^	SAS ^2^	
*CCR2*	chemokine receptor, participates in chemotaxis of monocytes	rs1799864	G/D	protein coding region, exon 1 (Val→Ile)	0.087	0.213	0.098	0.13
*IFIH1*	signaling molecule and intracellular viral RNA sensor	rs1990760	G/D	protein coding region, exon 15 (Ala→Thr)	0.605	0.187	0.564	0.58
*TNF*	pro-inflammatory cytokine	rs1800629	G/D	~300 nucleotides upstream 5′ UTR	0.134	0.059	0.053	0.13
*AC2*	cell receptor for SARS-CoV-2	rs2074192	G/D	intron	0.324	0.324	0.165	0.43
*MBL2*	protein binding to microorganisms and complement activator	rs1800450	G/D	protein coding region, exon 1 (Gly→Asp)	0.141	0.148	0.153	0.16
*CCL2*	chemokine, an activator of monocytes and basophils	rs1024611	A/G	~2500 nucleotides upstream 5′ UTR	0.316	0.547	0.321	0.3
*TRAF6*	signaling molecule, provides signal transmission from TNF receptors	rs4755453	G/C	intron	0.143	0.131	0.243	0.2
*TIRAP*	signaling molecule and activator of several kinase pathways	rs595209	T/G	intron	0.202	0.625	0.365	0.24
*CCL17*	cytokine, a chemotaxis activator of T-lymphocytes	rs223828	C/T	intron	0.04	0.3	0.078	0.08
*IFITM3*	signaling molecule, provides antiviral immunity	rs12252	T/C	protein coding region, exon 1 (synonymous mutation)	0.041	0.528	0.147	0.07
*CXCL8*	chemokine and the main mediator of inflammation	rs4073	A/T	~260 nucleotides upstream 5′ UTR	0.579	0.586	0.596	0.57
*SFTPD*	component of the pulmonary surfactant	rs721917	T/C	protein coding region, exon 1 (Met→Thr)	0.421	0.615	0.706	0.47
*HSPA1A*	heat shock protein	rs2227956	C/T	protein coding region, exon 2 (Thr→Met)	0.157	0.227	0.159	0.15
*TIPRSS2*	transmembrane serine protease	rs75603675	G/T	protein coding region, exon 1 (Gly→Asp)	0.405	0.017	0.222	0.41
*TLR2*	toll-like receptor, a pathogen recognition molecule	rs1898830	A/G	intron	0.325	0.433	0.585	0.41
		rs7656411	T/G	3′-untranslated region	0.272	0.481	0.253	0.22
*NOD2*	signaling molecule, recognizes bacterial liposaccharides	rs3135500	G/D	3′-untranslated region	0.418	0.176	0.179	0.41
*STAT3*	cytokine-activated transcriptional regulator of immune response genes	rs744166	T/C	intron	0.414	0.398	0.511	0.33
*STAT6*	cytokine-activated transcriptional regulator of immune response genes	rs324011	G/D	intron	0.347	0.253	0.337	0.39
*TLR9*	toll-like receptor, a pathogen recognition molecule	rs187084	T/C	~1400 nucleotides upstream 5′ UTR	0.427	0.405	0.37	0.4
		rs352162	C/T	~2100 nucleotides downstream of 3′ UTR	0.599	0.456	0.541	0.47
*C3AR1*	C3 receptor component complement, a chemotaxis activator	rs7842	A/G	3′-untranslated region		0.31 (HapMap)		0.29
*IFNL4*	Interferon	rs12979860	G/D	intron	0.309	0.08	0.233	0.33
*TLR7*	toll-like receptor, a pathogen recognition molecule	rs179008	A/T	protein coding region, exon 2 (Asp→Gly)	0.176	0	0.039	0.22
*TLR4*	toll-like receptor, a pathogen recognition molecule	rs4986790	A/G	protein coding region, exon 3 (Gln→Leu)	0.057	0	0.126	0.05
		rs4986791	C/T	protein coding region, exon 3 (Thr→Ile)	0.058	0	0.117	0.06
*TLR3*	toll-like receptor, a pathogen recognition molecule	rs3775291	C/T	protein coding region, exon 4 (Leu→Phe)	0.324	0.328	0.263	0.32
*IL1RN*	cytokine, a type 1 interleukin activity modulator	rs408392	C/G	intron	0.292	0.094	0.3	0.27
*IL13*	cytokine, a regulator of inflammation and maturation of B-cells	rs1800925	C/T	5′-untranslated region	0.178	0.178	0.2	0.27
*IL6*	cytokine, a regulator of maturation and differentiation of B-cells	rs1800796	G/C	5′-untranslated region	0.048	0.79	0.395	0.11
*TNFAIP3*	NFk-B inhibitor	rs6920220	G/D	200 K nucleotides upstream of the 5′-end gene (regulatory element)	0.169	0.004	0.098	0.19
*MYD88*	Signaling moleculeand an immunity adapter protein	rs7744	A/G	3′-untranslated region	0.144	0.33	0.099	0.17

^1^ The genotype is indicated for the DNA strand where the gene is located. First, the main allele is indicated, then the secondary allele. ^2^ EUR, data on the frequency of occurrence in the European population; EAS, for the population of East Asia; SAS, for the population of South Asia.

**Table 4 biomedicines-10-00549-t004:** Analysis of allele frequencies and genotype distribution with the Hardy–Weinberg equilibrium (HWE).

Gene	Polymorphism	Allele Frequency	Genotype Distribution	HWE *p*-Value
*TNF*	rs1800629	G 0.87/A 0.13	298/93/6 (0.75/0.23/0.02)	0.83
*TMPRSS2*	rs75603675	C 0.59/A 0.41	138/195/64 (0.35/0.49/0.16)	0.76
*STAT3*	rs744166	A 0.67/G 0.33	181/172/44 (0.46/0.43/0.11)	0.73
*STAT6*	rs324011	C 0.61/T 0.39	145/196/56 (0.37/0.49/0.14)	0.46
*CCR2*	rs1799864	G 0.87/A 0.13	301/87/9 (0.76/0.22/0.02)	0.38
*TLR7*	rs179008	A 0.78/T 0.22	278/60/59 (0.7/0.15/0.15)	**<0.0001**
*IFIH1*	rs1990760	T 0.58/C 0.42	141/179/77 (0.36/0.45/0.19)	0.15
*C3AR1*	rs7842	T 0.71/C 0.29	192/181/24 (0.48/0.46/0.06)	**0.037**
*TLR2*	rs1898830	A 0.59/G 0.41	125/216/56 (0.31/0.54/0.14)	**0.017**
*IFITM3*	rs12252	A 0.93/G 0.07	347/45/5 (0.87/0.11/0.01)	**0.031**

*p*-values for SNPs that deviated from the HWE are marked by bold font.

**Table 5 biomedicines-10-00549-t005:** Markers associated with the risk of mild or asymptomatic to more severe disease (subgroup A1 (healthy + asymptomatic + mild) vs. subgroup A2 (medium + severe)).

Gene	Polymorphism	Risk Genotype	OR	95% CI	*p*
*TNF*	rs1800629	GA + AA	1.98	1.23–3.18	0.0042
*TIPRSS2*	rs75603675	CA + AA	1.83	1.21–2.78	0.0043
*STAT3*	rs744166	GG	0.37	0.19–0.72	0.0025
*STAT6*	rs324011	TT	1.84	1.01–3.36	0.0041
*IFIG1*	rs1990760	CT + QC	1.57	1.04–2.37	0.033
		CC	1.75	1.04–2.95	0.032
*TLR7*	rs179008	GG	1.85	1.03–3.32	0.036

**Table 6 biomedicines-10-00549-t006:** Markers associated with the risk of the asymptomatic, mild, or moderate forms becoming severe (subgroup B1 (healthy + asymptomatic + mild + medium) versus subgroup B2 (severe)).

Gene	Polymorphism	Risk Genotype	OR	95% CI	*p*
*TNF*	rs1800629	GA + AA	1.88	1.18–2.97	0.0074
*TIPRSS2*	rs75603675	CA + AA	1.86	1.2–2.89	0.005
*STAT3*	rs744166	GG	0.36	0.17–0.78	0.0053
*IFIG1*	rs1990760	CC	1.80	1.09–2.98	0.021
*STAT6*	rs324011	TT	1.83	1.04–3.24	0.037

**Table 7 biomedicines-10-00549-t007:** Markers associated with susceptibility to disease or clinical manifestations of symptoms (subgroup C1 (healthy + asymptomatic patients) vs. subgroup C2 (symptomatic patients)).

Gene	Polymorphism	Risk Genotype	OR	95% CI	*p*
*KCR2*	rs1799864	GA + AA	2.21	1.12–4.39	0.015
*IFIG1*	rs1990760	CC	2.41	1.11–5.26	0.016
*TIPRSS2*	rs75603675	CA + AA	1.71	1.03–2.83	0.039
*S3AR1*	rs7842	TC + CC	2.08	1.25–3.46	0.0042
*STAT3*	rs744166	GG	0.39	0.15–0.57	0.0005

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
