# Peer review of "The Role of Genetic Factors in the Development of Acute Respiratory Viral Infection COVID-19: Predicting Severe Course and Outcomes"

_biomedicines, 2022, doi:10.3390/biomedicines10030549_

Round 1

Reviewer 1 Report

-Avoid the term foreign studies. I suggest to use data from other populations, or studies performed by other groups.

-The introduction is a mess of data about the candidate SNPs, that I think is  more appropiate for the discussion. Just mention the candidate gene with its putative role and how the variants might thus affect the risk.

-The main problem with this and other studies is the limited sample size, and in this study in particular due to the high number of markers. Moreover, the patients were stratified by severity in several groups with few patients.

-What about the statistical power?. When you design this type of study you need to know the minimum sample size to reach a power of 80. Or at least define the post hoc power with the patients you have studied.

-What about the Hardy-Weinberg in the study cohorts, patients and controls?. I suggest to show the genotypes in patients and controls in a table.

-Major revision: reduce the length of the introduction, as it is seems mor a revision than an original ms. Reduce the legth of the discussion, the results are more a proof of concept than a “hard” conclusion, mainly because with the sample size studied it is impossible to conclude a statisticaly powered association unless you are dealing with highly pathogenic variants. The supplementary data, in a separate file.

Author Response

Response to Reviewer 1 Comments

Point 1: Avoid the term foreign studies. I suggest to use data from other populations, or studies performed by other groups

Response 1: We replaced the term

Point 2: The introduction is a mess of data about the candidate SNPs, that I think is  more appropiate for the discussion. Just mention the candidate gene with its putative role and how the variants might thus affect the risk.

Response 2: We moved part of introduction to discussion and removed some sentences.

Point 3: The main problem with this and other studies is the limited sample size, and in this study in particular due to the high number of markers. Moreover, the patients were stratified by severity in several groups with few patients.

Response 3: Yes, it’s a global problem. But we believe that the tendence we have shown in our study is right. As a result of the statistical power analysis, data were obtained confirming the relative insufficiency of the sample size. However, data for rs1799864, rs1990760, rs1800629, rs75603675 and rs7842 actually are close to required N and may satisfy this criterion. As deal to patients’ stratification - we try to choose maximally objective criterion - severity of lung damage confirmed by CT scan.

Point 4: What about the statistical power?. When you design this type of study you need to know the minimum sample size to reach a power of 80. Or at least define the post hoc power with the patients you have studied.

Response 4: We added statistical power data to Supplementary materials and placed reference into the text.

Point 5: What about the Hardy-Weinberg in the study cohorts, patients and controls?. I suggest to show the genotypes in patients and controls in a table.

Response 5: We added brief table with HWE data of whole cohort to Result section and placed the table with full HWE data into Supplementary materials

Point 6: Major revision: reduce the length of the introduction, as it is seems mor a revision than an original ms. Reduce the legth of the discussion, the results are more a proof of concept than a “hard” conclusion, mainly because with the sample size studied it is impossible to conclude a statisticaly powered association unless you are dealing with highly pathogenic variants. The supplementary data, in a separate file.

Response 6: We reduced the length of introduction, but we have some difficulty in reducing of discussion. Would you be so kind to clarify us - wich moments in discussion are long or extra in your opinion.

Reviewer 2 Report

Dear Author,

This a very interesting scientific manuscript and it addresses a topic of high importance, that multigenetic polymorphism is associated with COVID-19 susceptibility and severity. Please, define all abbreviations and symbols. This MS requires significant English language editing (please see e.g., in Introduction).

Please, see new references:

  • Zepeda-Cervantes J, Martínez-Flores D, Ramírez-Jarquín JO, Tecalco-Cruz ÁC, Alavez-Pérez NS, Vaca L, Sarmiento-Silva Implications of the Immune Polymorphisms of the Host and the Genetic Variability of SARS-CoV-2 in the Development of COVID-19. . 2022 Jan 6;14(1):94.
  • Suh S, Lee S, Gym H, Yoon S, Park S, Cha J, Kwon DH, Yang Y, Jee SH. A systematic review on papers that study on Single Nucleotide Polymorphism that affects coronavirus 2019 severity. BMC Infect Dis. 2022 Jan 
  • Anna Malkova , Dmitriy Kudlay , Igor Kudryavtsev, Anna Starshinova , Piotr Yablonskiy and Yehuda Shoenfeld. Immunogenetic Predictors of Severe COVID-19 Vaccines 2021, 9, 211.

Issues that the author should consider in order to strengthen the manuscript are as follows:

Minor points:

KEYWORDS: antiviral immunity

INTRODUCTION:

Line 28-29; Severe acute respiratory syndrome coronavirus 2 ( SARS-CoV-2) not only leaves behind other viruses statistically in terms of virulence,  but also has a negative impact on economic factors due to  quarantine and stay-at- home regimes restrictions.

Further comments/suggestions:

Line 29-30; In addition, people of middle and mature age that make up the bulk of able-bodied people, including highly professional specialists, politicians  and medical workers contributing to the Gross Domestic Product (GDP) of the Russian Federation, are the group most susceptible to infection and morbidity

Line 33-37; The study of genetic polymorphisms (Single Nucleotide Polymorphism, SNP) that determine the manifestation of viral infection and corona virus disease 2019 (COVID-19) disease ought to be carried out in various categories of patients (explain please), both with a pronounced clinical performance and circulators (???) of SARS-CoV-2 viral infection confirmed by a real-time reverse transcription -polymerase chain reaction (RT-PCR) assay with real time detection or a positive SARS-CoV-2 antibody test.

Line 38-40; Maybe better say: One of the main complications of severe COVID-19 disease is cytokine release syndrome (CRS), also known the so-called  as a "cytokine storm", a pronounced systemic  increase of inflammatory  mediators  and abnormally high production of cytokines, chemokines and other factors of the inflammatory response [2].

Line 43-45; Maybe better say: It does not develop in all patients with Clinical variation in COVID-19  severity may also  be is the  result of the combination of a number of  multiple factors unique to each person´s his genotype and  phenotype depending on environmental factors.

Line 53-54; The study was carried out on genomic deoxyribonucleic acid (DNA) samples from patients with a confirmed  (RT-PCR test and/or positive COVID-19 antibody test) COVID-19 diagnosis

Line 54-62; An attempt was made to  identify genetic markers (single nucleotide polymorphisms) associated with the course of the disease, as well as the risk of its occurrence. Previous ly published foreign??? studies have shown that the main risk factors for severe course and high mortality in COVID-19 disease are: old age (>70 years), obesity, type 2 diabetes mellitus and arterial hypertension [3-6]. Clinical studies of COVID-19 patients showed that the most frequent concomitant diseases were arterial hypertension (56.6%), obesity (41.7%) and diabetes mellitus (33.8%) [7-8]. Previously, no such data were published for the Russian population in general and the Moscow population in particular (??? Please explain and add a ref).

Line 62-64; Our study confirms the conclusions of foreign colleagues: among patients with COVID-19, concomitant diagnoses of arterial hypertension (64.1%), obesity (30.9%) and type 2 diabetes mellitus (25.1%) prevailed (please add references).

Line 65-70; In this study, polymorphic markers in genes,  the activity of which was associated  with the COVID-19 pathogenesis, were analyzed. When creating the research protocol, modern data science obtained by research from various countries scientists from different countries in the process of studying imagerecognizing image-recognizing receptors and signaling pathways for immune response development  was used. When forming the list of markers, preference was given to genetic variants in the non-coding regions of genes (5'- and 3'-untranslated regions, as well as introns).

Line 71-76; The analyzed genes may be conditionally divided into the following groups: 1) genes  encoding the host’s cellular membrane receptor proteins for viral particles, 2) genes encoding proteins of signaling pathways of the host’s innate antiviral immuneity response (complement cascade activation,  production of interferons, activation of macrophages and lymphocytes) and 3) genes encoding pro-inflammatory cytokines, the overproduction of which causes the "cytokine storm".

Line 80-85; The first pathway is initiated by membrane Toll-like receptors (TLR)-2 and TLR4, which are image-recognition molecules against pathogens. It involves intracellular carrier TLR signaling molecules  carriers of signals received from toll-like receptors  - myeloid differentiation primary response 88 (MyD88), interleukine (IL)-1 receptor-associated kinase (IRAK), tumor necrosis factor (TNF) receptor-associated factor 6 (TRAF6), which in turn  activate the transcription factors  such as nuclear factor (NF)kB, activating protein-1 (AP-1), and interferon (IFN) regulatory factor (IRF5) [9]. This pathway ultimately leads  to the activation of genes of pro-inflammatory cytokines IL-1-beta, interleukin IL- 6, tumor necrosis factor (TNF) alpha, and ensures the development of early inflammatory reactions.

Line 87-90; The second signaling pathway begins with the recognition of viral nucleic acids by the intracellular toll-like receptors TLR3, TLR7, and TLR8. This pathway leads through  the chain of signaling molecules TRAF3, TANK-binding kinase 1 (TBK), and IRF3 to the activation of antiviral defense and a late inflammatory response due to the production of interferons IFNalpha and beta [9]. Moreover, TLR3 activation also triggers the MyD88 signaling pathway.

Line 91-94; In the case of COVID-19, both pathways associated with the activation of TLR2 and TLR3 are important playing a key critical role in the development of a timely immune response in patients with non-pathogenic polymorphisms in any of the genes involved in the synthesis of TLR-dependent signaling chain proteins.

Line 95-100; The third pathway is the complement activation pathway. Genes encoding components of this pathway were also included in the study. Finally, an important role in the pathogenesis of COVID-19 is played by the angiotensin-converting enzyme-2 (ACE-2), which is the main receptor ensuring the penetration of SARS-CoV-2 into the cell along with the transmembrane protease serine 2 (TMPRSS2) [10]. Polymorphisms in the ACE-2 gene play an important role in individual susceptibility to the coronavirus infection [11,21].

MATERIALS and METHODS:

I would recommend adding inclusions and exclusions criteria and also clinical characterization both of control group and COVID-19 diagnosed patients.

Line 143-146; The study included patients of the Intensive Care Units (ICU) of the N.V. Sklifosovsky Scientific Research Institute of First Aid, Clinical Hospital No. 1 of Medsi, City Clinical Hospital No. 40, as well as employees of AFK Sistema and Sistema BioTech Laboratory. The total sample size  was 397 persons, including 319 patients with a positive PCR test for SARS-CoV-2 (please describe if the samples were collected on different sites and describe three key steps you were using during sample collection) a  COVID-19 diagnosis (please explain the methods used for COVID-19 diagnosis).

Line 146-152; Maybe better say:According to the CT-SS, the patients were divided into three 3 subgroups: The first subgroup ( mild (99 persons patients): included asymptomatic circulators (no symptoms) and patients with mild pulmonary involvement  (CT-1); the second subgroup ( medium: 65 persons) included patients with medium moderate pulmonary involvement (CT-2),  and the third subgroup (severe: (155 persons ) included patients with severe (CT-3) and extremely very severe (CT-4) pulmonary involvement.

Line 153-154; The severity of lung involvement in COVID-19 was determined by chest computed tomography (CT) scan[12].  (Please explain in details the CT scan severity scoring system (CT-SS) and add the reference if the score is an adaptation of a method previously used.)

Line 155-158; The control group consisted of 78 persons who had contacts with patients, with negative results of regular RT-PCR testing for SARS-CoV-2 and the absence of antibodies to  SARS-CoV-2 - medical specialists of units for patients with COVID-19, and family members of patients who had COVID-19. Demographic data are presented in Tables 1 and 2, full complete phenotypic data and medical history of the patients are presented in Supplementary table 1S.

2.2. Genomic DNA isolation

Line 164-  EDTA-stabilized venous blood was used as a biological material for genomic DNA isolation.

(Please describe when the blood samples when collected and in which condition the samples were stored).  yDNA was isolated using a DiaGene whole blood DNA extraction kit (Dia-M, Russian Federation). The isolated DNA purity was checked using a NanoDrop OneC spectrophotometer (Themo FS, USA), the A260/280 ratio ranged from 1.8 to 1.91, and the A260/230  ratio ranged from 1.62 to 2.28. DNA concentration was measured using a dsDNA BR kit  on a Qubit Flex fluorometer (Thermo FS, USA). The concentration ranged from 15 to 300 ng/μl. The concentration of all DNA samples was adjusted to 2 ng/μl.

2.3. DNA genotyping

Please, describe a procedure for validation controls.

3.2. Statistical analysis.

Line 225: Thus, for all polymorphisms, three options (depending on the division of patients into groups) were made for calculating the statistical significance using two modelsdominant models: dominant and recessive.  

Line 243: The approach was implemented in the p.adjust() R function (adjust p-values for multiple comparisons?, please explain the method).

DISCUSSION:

The paper would be significantly improved by the addition of a more detailed comparison of clinical results vs experimental findings. It would also be a good idea to highlight the limitations of this study.

Line 273: According to information from key literature references for August 2020, foreign??? experts found that minor alleles of polymorphisms in the CCL2, IFIH1, MBL2, IFITM3 and other genes were associated with SARS-CoV, as well as with other viral respiratory infections [13-15].

Line 273: To date, the studies carried out do not provide a holistic understanding of 276 the molecular genetic mechanisms underlying a person's susceptibility to one or another 277 manifestation, outcome and or complications of COVID-19 disease due to insufficient data 278 and novelty of SARS-CoV-2.

Line 283: Proteins involved in the pathogenic pathways of COVID-19 (and the genes encoding  them) can be divided as follows, depending on their function.  1. Recognition of surface viral antigens (image-recognizing receptors, ACE-2)

4.1. TNF

Line 283: It disrupts the function of regulatory T cells (Treg) in people with rheumatoid arthritis.

Line 329-331; Saleh A. et al. tested 900 patients with COVID-19 infection and 184 people from the 329 control group (medical workers who had contact with patients with COVID-19 between 330 April and July 2020) for polymorphism of the TNF G-308A promoter [16].

Line 347-348:These  This data suggests that allele A is associated with more aggressive disease, and this  may be due to variations in serum TNF-α levels.  Ding et al. demonstrated associations between the A allele rs1800629 in TNF and the  G (C) rs1800796 allele in IL6 and a higher susceptibility to ARDS [17].

4.2. TMPRSS2

Line 382; But data obtained indicates that circulators of the minor T allele are more susceptible to infection with SARS-CoV-2 (OR = 1.71, 95% CI 1.03-2. 83, P = 0.039),  as well as the risk of developing a mild form of COVID-19 into a medium (OR = 1.83, 95%  CI 1.21-2.78, P = 0.0043) or severe one (OR = 1.86, 95% CI 1.2-2.89, P = 0.005).  

4.7. CCR2

Line 601-615; CCL2 is an important chemokine associated with the severity of COVID-19. It is activated at an early stage of an infectious disease and in the later stages in fatal cases its concentration rises sharply. In the lungs, CCL2 is mainly produced by alveolar macrophages, T lymphocytes, and endothelial cells, while its related receptor CCR2  is mainly expressed on monocytes and T lymphocytes [65]. In addition, the presence of  CCR2-bearing blood monocytes enhances neutrophil accumulation, dramatically reflect- 606 ing the cooperation and coordination between monocytes and neutrophils in leukocyte  efflux during pneumonia. In addition, CCL2 has been reported to increase procollagen  synthesis by fibroblasts. Together, these functions of CCL2 can lead to fibroproliferative complications in ARDS. One  review [65] was devoted to the place of chemokines and their receptors in the pathogenesis of COVID-19 and the relationship between the level of their expression and the severity and prognosis of the disease. Comparing the chemokine profile of SARS-CoV-2 with SARS-CoV and MERS-CoV, it is concluded that CXCL8, CXCL10 and CCL2 make a 614 decisive contribution to the pathogenesis of the disease in all three coronaviruses.

4.9. IFITM3

Line 692-700; It suppresses the penetration of viruses into the cytoplasm of the host cell, preventing the fusion of viruses with cholesterol-depleted endosomes. It can inactivate new viruses leaving the infected cell, allowing them to escape through the cholesterol-depleted  membrane. It is active against several viruses, including influenza A virus, SARS coronavirus (SARS-CoV), Marburg virus (MARV) and Ebola virus (EBOV), dengue virus (DNV), West Nile virus (WNV), Human immunodeficiency virus type 1 (HIV-1) and vesicular stomatitis virus (VSV). It may suppress: viral penetration mediated by influenza hemagglutinin protein, viral penetration mediated by GP1,2 MARV and EBOV, viral penetration mediated by SARS-CoV protein, and viral penetration mediated by VSV G protein.

  1. Conclusions

Line 726-735; In this work, an attempt was made for the first time to identify the genetic determinants of morbidity and risk of severe course for the Russian (Moscow) population of respiratory viral infections. Genetic polymorphic variants in genes, according to foreign literature data, involved in the pathogenesis of COVID-19 were investigated.

A group of statistically significant genetic variants associated with COVID-19 disease was identified. At the same time, it is interesting that almost all of these genetic variants are located in genes that are somehow responsible for innate (primary) immunity - CCR2, IFIH1, TMPRSS2, TNF, C3AR1, STAT3, STAT6, TLR2, and TLR7. Minor homozygous variants in the STAT3 and TLR2 genes are protective in terms of both the susceptibility and the risk of exacerbation of the disease and the severity of the course.

The course of the corona-virus epidemic has shown the need to predict morbidity, mortality and to assess the economic components and the structure of medical care during the epidemic.

Author Response

This a very interesting scientific manuscript and it addresses a topic of high importance, that multigenetic polymorphism is associated with COVID-19 susceptibility and severity. Please, define all abbreviations and symbols. This MS requires significant English language editing (please see e.g., in Introduction).

Please, see new references:

  • Zepeda-Cervantes J, Martínez-Flores D, Ramírez-Jarquín JO, Tecalco-Cruz ÁC, Alavez-Pérez NS, Vaca L, Sarmiento-Silva Implications of the Immune Polymorphisms of the Host and the Genetic Variability of SARS-CoV-2 in the Development of COVID-19. . 2022 Jan 6;14(1):94.
  • Suh S, Lee S, Gym H, Yoon S, Park S, Cha J, Kwon DH, Yang Y, Jee SH. A systematic review on papers that study on Single Nucleotide Polymorphism that affects coronavirus 2019 severity. BMC Infect Dis. 2022 Jan 
  • Anna Malkova , Dmitriy Kudlay , Igor Kudryavtsev, Anna Starshinova , Piotr Yablonskiy and Yehuda Shoenfeld. Immunogenetic Predictors of Severe COVID-19 Vaccines 20219, 211.

We added these references into introduction

Issues that the author should consider in order to strengthen the manuscript are as follows:

Minor points:

KEYWORDS: antiviral immunity

INTRODUCTION:

Line 28-29; Severe acute respiratory syndrome coronavirus 2 ( SARS-CoV-2) not only leaves behind other viruses statistically in terms of virulence,  but also has a negative impact on economic factors due to  quarantine and stay-at- home regimes restrictions.

Accepted

Further comments/suggestions:

Line 29-30; In addition, people of middle and mature age that make up the bulk of able-bodied people, including highly professional specialists, politicians  and medical workers contributing to the Gross Domestic Product (GDP) of the Russian Federation, are the group most susceptible to infection and morbidity

Accepted

Line 33-37; The study of genetic polymorphisms (Single Nucleotide Polymorphism, SNP) that determine the manifestation of viral infection and corona virus disease 2019 (COVID-19) disease ought to be carried out in various categories of patients (explain please) (different sex, age, concomitant disorders, etc), both with a pronounced clinical performance and circulators (???)asymptomatic carriers of SARS-CoV-2 viral infection confirmed by a real-time reverse transcription -polymerase chain reaction (RT-PCR) assay with real time detection or a positive SARS-CoV-2 antibody test.

Accepted

Line 38-40; Maybe better say: One of the main complications of severe COVID-19 disease is cytokine release syndrome (CRS), also known the so-called  as a "cytokine storm", a pronounced systemic  increase of inflammatory  mediators  and abnormally high production of cytokines, chemokines and other factors of the inflammatory response [2].

We decided to remove any mentions about cytokine storm due to recommendations of other reviewers

Line 43-45; Maybe better say: It does not develop in all patients with Clinical variation in COVID-19  severity may also  be is the  result of the combination of a number of  multiple factors unique to each person´s his genotype and  phenotype depending on environmental factors.

Accepted

Line 53-54; The study was carried out on genomic deoxyribonucleic acid (DNA) samples from patients with a confirmed  (RT-PCR test and/or positive COVID-19 antibody test) COVID-19 diagnosis

DNA is common abbreviation and we have never met its explanation 

Line 54-62; An attempt was made to  identify genetic markers (single nucleotide polymorphisms) associated with the course of the disease, as well as the risk of its occurrence. Previous ly published foreign??? studies replaced by "data from other populations" have shown that the main risk factors for severe course and high mortality in COVID-19 disease are: old age (>70 years), obesity, type 2 diabetes mellitus and arterial hypertension [3-6]. Clinical studies of COVID-19 patients showed that the most frequent concomitant diseases were arterial hypertension (56.6%), obesity (41.7%) and diabetes mellitus (33.8%) [7-8]. Previously, no such data were published for the Russian population in general and the Moscow population in particular (??? Please explain and add a ref).

At the time of manuscript writing there were no data about genetic determinants of COVID-19 severity for the Russian population

Line 62-64; Our study confirms the conclusions of foreign colleagues: among patients with COVID-19, concomitant diagnoses of arterial hypertension (64.1%), obesity (30.9%) and type 2 diabetes mellitus (25.1%) prevailed (please add references).

We removed this sentence due to requirements of other reviewers

Line 65-70; In this study, polymorphic markers in genes,  the activity of which was associated  with the COVID-19 pathogenesis, were analyzed. When creating the research protocol, modern data science obtained by research from various countries scientists from different countries in the process of studying imagerecognizing image-recognizing receptors and signaling pathways for immune response development  was used. When forming the list of markers, preference was given to genetic variants in the non-coding regions of genes (5'- and 3'-untranslated regions, as well as introns).

Accepted

Line 71-76; The analyzed genes may be conditionally divided into the following groups: 1) genes  encoding the host’s cellular membrane receptor proteins for viral particles, 2) genes encoding proteins of signaling pathways of the host’s innate antiviral immuneity response (complement cascade activation,  production of interferons, activation of macrophages and lymphocytes) and 3) genes encoding pro-inflammatory cytokines, the overproduction of which causes the "cytokine storm".

Line 80-85; The first pathway is initiated by membrane Toll-like receptors (TLR)-2 and TLR4, which are image-recognition molecules against pathogens. It involves intracellular carrier TLR signaling molecules  carriers of signals received from toll-like receptors  - myeloid differentiation primary response 88 (MyD88), interleukine (IL)-1 receptor-associated kinase (IRAK), tumor necrosis factor (TNF) receptor-associated factor 6 (TRAF6), which in turn  activate the transcription factors  such as nuclear factor (NF)kB, activating protein-1 (AP-1), and interferon (IFN) regulatory factor (IRF5) [9]. This pathway ultimately leads  to the activation of genes of pro-inflammatory cytokines IL-1-beta, interleukin IL- 6, tumor necrosis factor (TNF) alpha, and ensures the development of early inflammatory reactions.

Line 87-90; The second signaling pathway begins with the recognition of viral nucleic acids by the intracellular toll-like receptors TLR3, TLR7, and TLR8. This pathway leads through  the chain of signaling molecules TRAF3, TANK-binding kinase 1 (TBK), and IRF3 to the activation of antiviral defense and a late inflammatory response due to the production of interferons IFNalpha and beta [9]. Moreover, TLR3 activation also triggers the MyD88 signaling pathway.

Line 91-94; In the case of COVID-19, both pathways associated with the activation of TLR2 and TLR3 are important playing a key critical role in the development of a timely immune response in patients with non-pathogenic polymorphisms in any of the genes involved in the synthesis of TLR-dependent signaling chain proteins.

Line 95-100; The third pathway is the complement activation pathway. Genes encoding components of this pathway were also included in the study. Finally, an important role in the pathogenesis of COVID-19 is played by the angiotensin-converting enzyme-2 (ACE-2), which is the main receptor ensuring the penetration of SARS-CoV-2 into the cell along with the transmembrane protease serine 2 (TMPRSS2) [10]. Polymorphisms in the ACE-2 gene play an important role in individual susceptibility to the coronavirus infection [11,21].

This part of introduction (lines from 71) was moved to discussion due to requirements of other reviewers.

MATERIALS and METHODS:

I would recommend adding inclusions and exclusions criteria and also clinical characterization both of control group and COVID-19 diagnosed patients.

Line 143-146; The study included patients of the Intensive Care Units (ICU) of the N.V. Sklifosovsky Scientific Research Institute of First Aid, Clinical Hospital No. 1 of Medsi, City Clinical Hospital No. 40, as well as employees of AFK Sistema and Sistema BioTech Laboratory. The total sample size  was 397 persons, including 319 patients with a positive PCR test for SARS-CoV-2 (please describe if the samples were collected on different sites and describe three key steps you were using during sample collection) a  COVID-19 diagnosis (please explain the methods used for COVID-19 diagnosis).

Inclusion criterion is positive PCR test for patient group and negative PCR test with combination of absence of COVID-19 antibodies for control group. Is it enough to say "PCR tests were carried out on RNA extarcted from oropharyngeal swabs" or additional information is necessary?

Line 146-152; Maybe better say:According to the CT-SS, the patients were divided into three 3 subgroups: The first subgroup ( mild (99 persons patients): included asymptomatic circulators (no symptoms) and patients with mild pulmonary involvement  (CT-1); the second subgroup ( medium: 65 persons) included patients with medium moderate pulmonary involvement (CT-2),  and the third subgroup (severe: (155 persons ) included patients with severe (CT-3) and extremely very severe (CT-4) pulmonary involvement.

Accepted

Line 153-154; The severity of lung involvement in COVID-19 was determined by chest computed tomography (CT) scan[12].  (Please explain in details the CT scan severity scoring system (CT-SS) and add the reference if the score is an adaptation of a method previously used.)

This method (including score system) is described in ref 12. We could insert this description, but it will take up much space

Line 155-158; The control group consisted of 78 persons who had long term contacts with patiheseents, with negative results of regular RT-PCR testing for SARS-CoV-2 and the absence of antibodies to  SARS-CoV-2 - medical specialists of units for patients with COVID-19, and family members of patients who had COVID-19. Demographic data are presented in Tables 1 and 2, full complete phenotypic data and medical history of the patients are presented in Supplementary table 1S.

Accepted

2.2. Genomic DNA isolation

Line 164-  EDTA-stabilized venous blood was used as a biological material for genomic DNA isolation.

(Please describe when the blood samples when collected and in which condition the samples were stored).  yDNA was isolated using a DiaGene whole blood DNA extraction kit (Dia-M, Russian Federation). The isolated DNA purity was checked using a NanoDrop OneC spectrophotometer (Themo FS, USA), the A260/280 ratio ranged from 1.8 to 1.91, and the A260/230  ratio ranged from 1.62 to 2.28. DNA concentration was measured using a dsDNA BR kit  on a Qubit Flex fluorometer (Thermo FS, USA). The concentration ranged from 15 to 300 ng/μl. The concentration of all DNA samples was adjusted to 2 ng/μl.

DNA samples were collected at the time of patient admission to the hospital or, for mild or asymptomatic patients at the time of oropharyngeal swab taking. But we think that's insignificant information due to lack of it's influence on genotyping. Collected samples were frozen at -20oC and processed in 2 days

2.3. DNA genotyping

Please, describe a procedure for validation controls.

At first, we determined polymorphism genotypes by qPCR with TaqMan probes. Then we took some samples with different genotypes of each analyzed SNP (2-3 samples with major homozygote genotype, 2-3 heterozygote and 2-3 minor homozygote) and analyzed their sequence by Sanger method at 3500 genetic analyzer to confirm the correspondence between genotype and fluorescence curves pattern. Primers for Sanger sequencing were designed using Primer-BLAST tool, PCR fragments had the length about 300 bp and included corresponding SNP area. In some cases, it was impossible to use TaqMan probes due to features of DNA around SNP, these polymorphisms were analyzed by Sanger sequencing. But none of these SNP did not show any significance and were not concidered further

3.2. Statistical analysis.

Line 225: Thus, for all polymorphisms, three options (depending on the division of patients into groups) were made for calculating the statistical significance using two modelsdominant models: dominant and recessive.  

Accepted

Line 243: The approach was implemented in the p.adjust() R function (adjust p-values for multiple comparisons?, please explain the method).

This sentence was removed due to requirements of another reviewer

DISCUSSION:

The paper would be significantly improved by the addition of a more detailed comparison of clinical results vs experimental findings. It would also be a good idea to highlight the limitations of this study.

OK, we try to do it

Line 273: According to information from key literature references for August 2020, foreign??? experts other scientists found that minor alleles of polymorphisms in the CCL2, IFIH1, MBL2, IFITM3 and other genes were associated with SARS-CoV, as well as with other viral respiratory infections [13-15].

Line 273: To date, the studies carried out do not provide a holistic understanding of 276 the molecular genetic mechanisms underlying a person's susceptibility to one or another 277 manifestation, outcome and or complications of COVID-19 disease due to insufficient data 278 and novelty of SARS-CoV-2.

Accepted

Line 283: Proteins involved in the pathogenic pathways of COVID-19 (and the genes encoding  them) can be divided as follows, depending on their function.  1. Recognition of surface viral antigens (image-recognizing receptors, ACE-2)

Accepted

4.1. TNF

Line 283: It disrupts the function of regulatory T cells (Treg) in people with rheumatoid arthritis.

Sorry, but we can't find this place in manuscript

Line 329-331; Saleh A. et al. tested 900 patients with COVID-19 infection and 184 people from the 329 control group (medical workers who had contact with patients with COVID-19 between 330 April and July 2020) for polymorphism of the TNF G-308A promoter [16].

Accepted

Line 347-348:These  This data suggests that allele A is associated with more aggressive disease, and this  may be due to variations in serum TNF-α levels.  Ding et al. demonstrated associations between the A allele rs1800629 in TNF and the  G (C) rs1800796 allele in IL6 and a higher susceptibility to ARDS [17].

Accepted

4.2. TMPRSS2

Line 382; But data obtained indicates that circulators of the minor T allele are more susceptible to infection with SARS-CoV-2 (OR = 1.71, 95% CI 1.03-2. 83, P = 0.039),  as well as the risk of developing a mild form of COVID-19 into medium (OR = 1.83, 95%  CI 1.21-2.78, P = 0.0043) or severe one (OR = 1.86, 95% CI 1.2-2.89, P = 0.005).  

4.7. CCR2

Line 601-615; CCL2 is an important chemokine associated with the severity of COVID-19. Iis activated at an early stage of an infectious disease and in the later stages in fatal cases its concentration rises sharply. In the lungs, CCL2 is mainly produced by alveolar macrophages, T lymphocytes, and endothelial cells, while its related receptor CCR2  is mainly expressed on monocytes and T lymphocytes [65]. In addition, the presence of  CCR2-bearing blood monocytes enhances neutrophil accumulation, dramatically reflect- 606 ing the cooperation and coordination between monocytes and neutrophils in leukocyte  efflux during pneumonia. In addition, CCL2 has been reported to increase procollagen  synthesis by fibroblasts. Together, these functions of CCL2 can lead to fibroproliferative complications in ARDS. One  review [65] was devoted to the place of chemokines and their receptors in the pathogenesis of COVID-19 and the relationship between the level of their expression and the severity and prognosis of the disease. Comparing the chemokine profile of SARS-CoV-2 with SARS-CoV and MERS-CoV, it is concluded that CXCL8, CXCL10 and CCL2 make a 614 decisive contribution to the pathogenesis of the disease in all three coronaviruses.

Accepted

4.9. IFITM3

Line 692-700; It suppresses the penetration of viruses into the cytoplasm of the host cell, preventing the fusion of viruses with cholesterol-depleted endosomes. It can inactivate new viruses leaving the infected cell, allowing them to escape through the cholesterol-depleted  membrane. It is active against several viruses, including influenza A virus, SARS coronavirus (SARS-CoV), Marburg virus (MARV) and Ebola virus (EBOV), dengue virus (DNV), West Nile virus (WNV), Human immunodeficiency virus type 1 (HIV-1) and vesicular stomatitis virus (VSV). It may suppress: viral penetration mediated by influenza hemagglutinin protein, viral penetration mediated by GP1,2 MARV and EBOV, viral penetration mediated by SARS-CoV protein, and viral penetration mediated by VSV G protein.

Accepted

  1. Conclusions

Line 726-735; In this work, an attempt was made for the first time to identify the genetic determinants of morbidity and risk of severe course for the Russian (Moscow) population of respiratory viral infections. Genetic polymorphic variants in genes, according to foreign literature data, involved in the pathogenesis of COVID-19 were investigated.

Accepted

A group of statistically significant genetic variants associated with COVID-19 disease was identified. At the same time, it is interesting that almost all of these genetic variants are located in genes that are somehow responsible for innate (primary) immunity - CCR2, IFIH1, TMPRSS2, TNF, C3AR1, STAT3, STAT6, TLR2, and TLR7. Minor homozygous variants in the STAT3 and TLR2 genes are protective in terms of both the susceptibility and the risk of exacerbation of the disease and the severity of the course.

Accepted

The course of the corona-virus epidemic has shown the need to predict morbidity, mortality and to assess the economic components and the structure of medical care during the epidemic.

Were should we place this sentence?

Reviewer 3 Report

Dear Authors,

I have read the manuscript and I send you my comments:

1) Methods: Ethic committee data and protocol must be added ; please add end points, inclusion and esclusion criteria, as well as experimental protocol

2) The paper is very long please add tables for each section in order to reduce the word in the section

Author Response

Greetings!
I reformatted the Results and Discussion sections. The discussion is somewhat abbreviated. If the size of the article is still too large - please indicate the maximum number of characters (with or without spaces).

With gratitude and respect.

Round 2

Reviewer 2 Report

Dear Author,

I appreciate that the Author have addressed my initial comments!

Reviewer 3 Report

Dear Auhtors,

I have read the manuscript and I have not comments

best regards